



# Variability and trend analysis of temperature in the upper troposphere and stratosphere region over the tropics (Réunion), by combining balloon-sonde and satellite measurements

Gregori de Arruda Moreira[1], Hassan Bencherif[2], Tristan Millet[2], Damaris Kirsch Pinheiro[3]

[1]Federal Institute of Education, Science and Techonology of São Paulo (IFSP), São Paulo, 01109-010, Brazil
[2]Laboratoire de l'Atmosphère et des Cyclones (LACy, UMR 8105 CNRS, Université de La Réunion, Météo-France), Reunion Island, France
[3]Federal University of Santa Maria, Santa Maria, 97105-900, Brazil

*Correspondence to*: Gregori de Arruda Moreira (gregori.moreira@ifsp.edu.br)

**Abstract.** Tropopause height and temperature play a crucial role in atmospheric chemistry and radiative forcing and serve as key indicators of anthropogenic climate change. However, accurately determining this parameter requires advanced remote sensing techniques. This study compares tropopause height estimates from in-situ and remote sensing instruments (SHADOZ and COSMIC-1) with reanalysis data from MERRA-2 over Réunion from 2006 to 2020. The results reveal strong agreement between vertical temperature profiles obtained from SHADOZ and COSMIC-1, demonstrating that both can reliably estimate tropopause height using the Cold Point Temperature (CPT) and/or Lapse Rate Temperature (LRT) methods. Conversely, while MERRA-2 assimilates data from these sources, its fixed vertical resolution limits its ability to capture tropopause height variations accurately. Given the consistency between SHADOZ and COSMIC-1, their data were combined to construct a more refined dataset, which was then used to assess temperature trends. The analysis indicates a high influence of annual and semi-annual oscillations in Tropopause height dynamics, as well as, a decreasing trend in CPT and a slight increase in the Lapse Rate Tropopause (LRT) height.

## 1 Introduction

The troposphere is the atmospheric layer extending from the Earth's surface to around 18 km, depending on the latitude, so it is generally higher at the equator and decreases toward the poles. Inside this layer can be found the Tropical Tropopause Layer (TTL), which extends from 12-14 to 18 km, and it represents the transition region between the well-mixed convective troposphere and the radiatively controlled stratosphere (Fueglistaler et al., 2009; Randel and Jensen, 2013). The TTL is the main gateway for air to enter the stratosphere and a crucial indicator of anthropogenic climate change, as indicated by variations in the height and/or temperature of the tropopause (Astudillo et al., 2014; Santer et al., 2004).

The tropopause, found within the TTL, is a significant physical boundary that separates the unstable and moist troposphere from the stable and dry stratosphere. The temperature and height of such layer are influenced by both tropospheric and

stratospheric forcing, like as changes in solar radiation (Reid and Gage, 1981), atmospheric angular momentum (Reid and
Gage, 1984), El Ninõ-Southern Oscillation (ENSO), stratospheric ozone (Dameris et al., 1995; Hoinka, 1998), Quasi-Biennal
Oscillation (QBO) variability, explosive volcanic eruptions (Reid and Gage, 1985; Randel et al., 2000), and concentrations of
Greenhouse Gases (GHG) (Zou et al., 2023). Consequently, tropospheric warming or stratospheric cooling can result in an
increase in the tropopause height (Astudillo et al., 2014; Santer et al., 2004).

Traditional methods for sounding tropopause structure are usually based on in situ measurements (e.g., weather stations,
radiosonde), model data (e.g., reanalysis data) and remote sensing (e.g., lidar, airborne, satellite soundings). The direct
sounding technique usually has an uneven global distribution, with only sparse data distribution, especially in the southern
hemisphere (Santer et al.,2004). On the other hand, reanalyses provide global and temporal coverage and uniform data type.
However, reanalyses suffer from coarser vertical resolution, which can render tropopause height detection unfeasible (Birner
et. al. 2006). Considering remote sensing, Global Navigational Satellite System Radio Occultation (GNSS-RO) stand out for
offering accurate tropospheric profiles with high vertical resolution and global coverage independently of weather conditions.

In this context, this study compares vertical temperature profiles from the Southern Additional Ozonesondes (SHADOZ)
network, Constellation Observing System for Meteorology Ionosphere and Climate 1 (COSMIC-1), and Modern Era
Retrospective analysis for Research and Applications – Version 2 (MERRA-2) over Réunion (2006–2020) to identify
similarities and/or differences. Additionally, the study demonstrates how these datasets contribute to understanding variability
in the Upper Troposphere–Lower Stratosphere (UT-LS) region.  Finally, by combining ground-based balloon-sonde data and
satellite-based COSMIC-1 observations, the variability and trend estimates of temperature in the tropical UT-LS region were
investigated.

The paper structure is as follows: Section 2 gives a brief description of the experimental site and instruments used. The
methodologies applied are presented in section 3. The comparisons between temperatures and tropopause height estimated
from SHADOZ, COSMIC-1 and MERRA-2 datasets are presented in section 4. In section 5, the results provided by the Trend-
Run model are described. Finally, the conclusions are given in Section 6.

## 2 Materials

### 2.1 Study Area

Réunion (21.10° S; 55.48° E) is a volcanic island of the Indian Ocean with a population of ∼860,000 inhabitants. It covers
around 2,512 km², and it is characterized by a humid tropical climate tempered by the oceanic influence of the trade winds
blowing from the southeast. In addition, this climate is endowed with great variability, mainly due to the island landscape,
which causes numerous microclimates (Britannica, 2025).



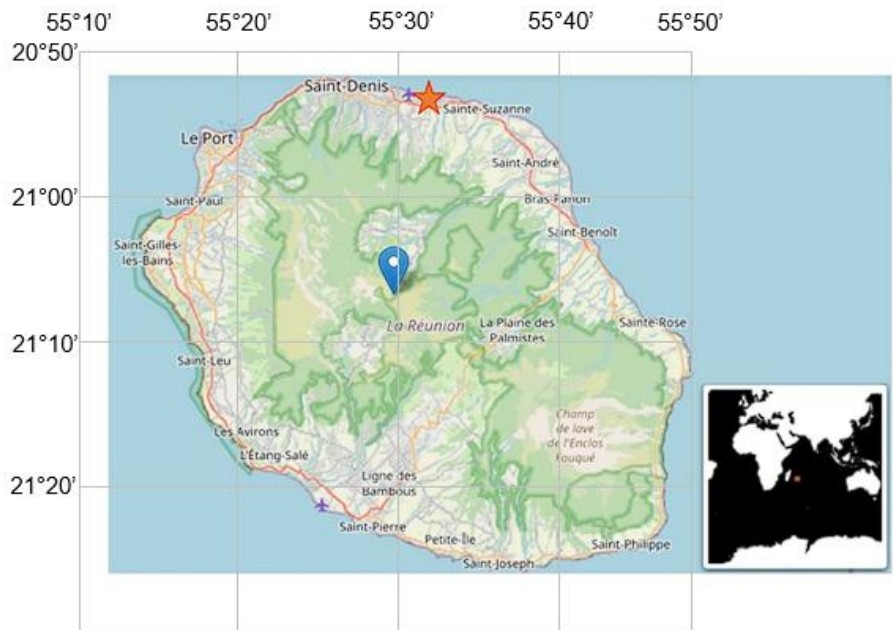

**Figure 1: Geographical location of the study site, Réunion (21,10° S; 55,48° E), a French overseas department in the southern tropics. The red star symbol indicates the measurement site where the balloon radiosondes are carried out, located to the north of the island at Roland Garros International Airport (20,9° S; 55,5° E).**

## 2.2 Measurement by ballon-sonde experiment

Radiosonde measurements began in Réunion in 1993 (Baldy et al., 1996), and the station joined the Southern Hemisphere ADditional OZonesondes (SHADOZ) network in 1998, increasing the frequency of measurements to weekly. The SHADOZ network is a NASA project that aims to fill gaps in ozone observation in the southern tropics by increasing radiosonde frequencies at existing stations on a cost-sharing basis (Thompson et al., 2003). Each radiosonde is coupled in a balloon meteorological sonde, with an electrochemical cell (ECC) ozone-sonde, to transmit in-situ information on air pressure, temperature, relative humidity, and ozone partial pressure. For this study, we used radiosonde temperature profiles from the study site (2006–2020), covering the COSMIC-1 mission's operational period and overlapping with its measurements. Radiosonde data can be accessed at the following link on the SHADOZ website (https://tropo.gsfc.nasa.gov/shadoz/Reunion.html).

## 2.2 Temperature profiles from the COSMIC-1 experiment

The Constellation Observing System for Meteorology, Ionosphere, and Climate 1 (COSMIC-1) is a joint U.S.-Taiwanese program designed to provide advances in meteorology, ionospheric research, climatology, and space weather by using GNSS-RO, which is a satellite remote sensing technique that uses GNSS (e.g. Global Positioning System - GPS) measurements



received by low-Earth orbiting satellites to profile the Earth's atmosphere and ionosphere with high vertical resolution and global coverage (Thompson et al., 2003, Cheng et al., 2006; Anthes et al., 2008). The COSMIC-1 was launched into a circular low-Earth orbit on April 15, 2006, and retired in 2020. In this paper, we utilized all temperature profiles that have been provided

by COSMIC-1 between 2006 and 2020. The profiles were selected for a specific geographic area, covering Réunion with a margin of ± 2° in latitude and ± 3° in longitude.

## 2.2 Temperature assimilation from the MERRA-2 reanalysis

The Modern-Era Retrospective Analysis for Research and Applications – Version 2 (MERRA-2) is a global atmospheric reanalysis produced by the National Aeronautics and Space Administration (NASA) Global Modeling and Assimilation Office

(GMAO). MERRA-2 replaces the original MERRA so that more information is assimilated, such as modern hyperspectral radiance and microwave observations, along with GPS-Radio Occultation and NASA ozone datasets. Like COSMIC-1 time cover, we used vertical temperature profiles from MERRA-2 data from 2006 to 2020. Such temperature profiles can be obtained         from         the         following         MERRA-2         product,         M2T3NVASM_5.12.4 (https://disc.gsfc.nasa.gov/datasets/M2T3NVASM_5.12.4/summary), which is composed of 17 atmospheric variables and has

a spatial resolution of 0.5 ° x 0.625°. All data are distributed on 72 pressure levels, ranging from 1000 to 0.01 hPa, with a time resolution of 3 hours. In this study, we used MERRA-2 average daily temperature profiles.

## 3 Methods

This study is based on the combination of 3 sources of temperature profile data at Réunion (radiosonde, COSMIC-1 and MERRA-2), over 15 years (2006-2020). Such a combination makes it possible to describe the thermal structure of the tropical

atmosphere from the ground up to the mesosphere. However, given the differences in the methods of acquisition and production, the temperature data used may not offer the same uncertainties, or resolutions everywhere at all altitudes and in all seasons. The first step is to compare these datasets to each other and define the altitude ranges where they can be used with confidence. Secondly, one could use all or a combination of these data to construct a coherent and regular time series, to investigate variability and temperature change in different atmospheric layers. Therefore, temperature gradients in the

atmosphere are crucial as they control most processes in the atmosphere such as thermodynamics, dynamics, and chemistry. They are critical for weather and climate forecasting. Regarding the values obtained from the methods described in the following subsections, all reported uncertainties represent ± 2σ variability unless stated otherwise.

## 3.1 Cold-Point Tropopause

Based on thermal properties, it is possible to use the temperature profile to estimate the tropopause height from the Cold-Point

Tropopause ($TH_{CPT}$), which can be defined as the height where the tropospheric temperature reaches its minimum value



(Selkrik, 1993). In this paper, the temperature observed at this height z is denominated $T_{CPT}$. Such definition is given by the following equation:

$$T(z) \begin{cases} T_{CPT} = \min\big(T(z)\big) \\ \quad TH_{CPT} = z, \\ where\ T(z) = T_{CPT} \end{cases} \tag{1}$$

## 3.2 Lapse-Rate Tropopause

From temperature profiles $T(z)$ it is possible to identify the tropopause height from the lowest level at which the lapse rate is less than 2 K.km$^{-1}$ and it remains within this level for the next 2 km, such a point is denominated as the Lapse-Rate Tropopause ($TH_{LRT}$) (WMO,1957), which is endowed of high stability, and signifies the thermodynamical transition between the troposphere and stratosphere. In this paper, the temperature detected at such height is denominated $T_{LRT}$. In addition, in all cases of the final time series where the criterium established in the previous paragraph was not identified, the $TH_{LRT}$ was

classified as NaN.

## 3. 3 Stratopause Height

As defined by France et al. (2012), the stratopause is determined as the altitude where the stratospheric temperature reaches its maximum in the vicinity of 50 km. Therefore, considering the 3 databases applied in this study, only the COSMIC-1 temperature profiles can be used to detect the stratopause, due to their vertical range and resolution.

**3. 4 Forcings parametrization in the Trend-Run model**

The Trend-Run model, a multiple linear regression model, was first adapted at the University of Réunion to analyze temperature trends in the southern subtropical upper troposphere-lower stratosphere (UT-LS) (Bencherif et al., 2006). This model decomposes the variations of a time-series signal, $S(t)$, into components representing atmospheric forcings:

$$S(t) = c_1 SAO(t) + c_2 AO(t) + c_3 QBO(t) + c_4 ENSO(t) + c_5 SSN(t) + c_6 IOD(t) + \varepsilon \tag{2}$$

where ε represents the residuals term which includes the trend, and $c_i$ (i ranging from 1 to 6) are contribution coefficients of the respective forcings. The coefficients $c_i$ can be derived using the least-squares method, which minimizes the residual variance, while the trend is parameterized as linear: $Trend(t) = a_0 + a_1 t$, where $a_0$ is a constant, and $a_1$ is the trend slope.

The initial model incorporated key forcings, including annual and semi-annual oscillations (AO, SAO), quasi-biennial oscillation (QBO), El Niño-Southern Oscillation (ENSO), sunspot numbers (SSN), AO and SAO represent seasonal cycles, with SAO being particularly dominant in the tropics above 35 km altitude. SAO amplitudes decrease with latitude but can intensify in the subtropics, depending on altitude. QBO is parametrised as a proxy of the zonal wind at 70 hPa derived from



balloon measurements at the Equator (Singapore), while the ENSO is parametrized with the Multivariate ENSO Index (Randel
and Cobb, 1994; Li et al., 2008). The model was further extended by Bègue et al. (2010) to include the Indian Ocean Dipole
(IOD), which describes sea surface temperature (SST) anomalies in the Indian Ocean's east-west dipole (Saji et al., 1999;
Morioka et al., 2010). The IOD, measured by the Dipole Mode Index (DMI), quantifies the SST difference between the western
(50°E–70°E, 10°S–10°N) and eastern (90°E–110°E, 10°S–Equator) Indian Ocean (Sivakumar et al., 2017). The DMI data
were sourced from the Japanese Agency for Marine-Earth Science and Technology. The model's performance, measured with
the coefficient of determination ($R^2$), evaluates its ability to explain how much the forcings explain the signal variability.
Decadal temperature trends (in Kelvin) were calculated to evaluate long-term changes. For details on the Trend-Run model
and its parameterizations, see Bencherif et al. (2006) and Bègue et al. (2010).

## 4 Analysis of temperature profiles

In this section is performed a comparison among the vertical temperature profiles provided by MERRA-2 ($T_{MERRA-2}(z)$),
SHADOZ ($T_{SHADOZ}(z)$) and COSMIC-1 ($T_{COSMIC-1}(z)$), The limitations and advantages of each system will be presented and
then a database will be created combining the data from the instruments that are endowed of the most similar profiles.

### 4. 1 Day-to-day Comparisons

Figure 2 shows daily comparisons of vertical temperature profiles from SHADOZ ($T_{SHADOZ}(z)$), COSMIC-1 ($T_{COSMIC-1}(z)$),
and MERRA-2 ($T_{MERRA-2}(z)$) for three selected dates: 25 June 2014 (Fig. 2a), 19 November 2014 (Fig. 2b), and 17 September
2014 (Fig. 2c). The corresponding temperature differences (SHADOZ − COSMIC-1 and SHADOZ − MERRA-2) for the same
dates are presented in Figures 2d, 2e, and 2f.





**Figure 2: Comparison between $T_{SHADOZ}(z)$ (red), $T_{COSMIC-1}(z)$ (green) and $T_{MERRA-2}(z)$ (blue) profiles on 25-06-2014 (a), 19-11-2014 (b) and 17-09-2014 (c) and the difference between $T_{SHADOZ}(z)$ and $T_{COSMIC-1}(z)$ (black line) and $T_{SHADOZ}(z)$ and $T_{MERRA-2}(z)$ (orange line) profiles to the same days (d), (e), and (f), respectively.**



On 25 June and 19 November, the difference between $T_{SHADOZ}(z)$ and $T_{MERRA-2}(z)$ does not exceed 4.0 K, so that the minimal differences are observed below 6.0 km and in the region between 16.0 km and 18.0 km. On the other hand, $T_{SHADOZ}(z)$ and $T_{COSMIC-1}(z)$ present a significant difference below 5.0 km (in some points the difference is higher than 40.0 K). However, above 10.0 km, on both days, $T_{SHADOZ}(z)$ and $T_{COSMIC-1}(z)$ are very similar, so that the difference does not exceed 5.0 K.

On 17 September, $T_{SHADOZ}(z)$ and $T_{MERRA-2}(z)$ present a difference lower than 5.0 K in the region below 15.0 km. Above 15.0 km, the difference increases significantly, mainly in the region between 15.0 and 20.0 km. As observed in the other two days, the higher difference between $T_{SHADOZ}(z)$ and $T_{COSMIC-1}(z)$ is observed in the first 5.0 km. However, a significant difference (around 6 K) also is observed in the region between 15.0 and 20.0 km, resulting in a variation of approximately 2.1 km between the CPT estimated by $T_{SHADOZ}(z)$ (17.3 km) and $T_{COSMIC-1}(z)$ (15.2 km). In addition, a difference between

$T_{COSMIC-1}(z)$ and $T_{MERRA-2}(z)$ in the region above 30.0 km is significantly higher than that observed in previous analyses. However, as will be demonstrated in the next sections, it is possible to consider 17 September as an exceptional case, where $T_{COSMIC-1}(z)$ presents an abnormal behavior.

## 4. 2 Weekly and daily temperature profiles

Figure 3 presents the curtain plot from the ground up to 30.0 km of weekly temperature profiles as measured by balloon-sondes
($T_{SHADOZ}$) at Réunion from 2006 to 2020. The figure shows that during this period, balloon-sonde measurements at Réunion were carried out almost continuously, at the rate of one release per week, except in 2007, due to a shortage of stock supplies, which resulted in a 3-month interruption. For each profile, the tropopause height was determined and corresponds to the location of the LRT ($LRT_{SHADOZ}$) and CPT ($CPT_{SHADOZ}$), as defined above. Their respective positions are superimposed in Figure 3 by grey and black dots. Overall, the altitude of the $CPT_{SHADOZ}$ varies between 16.0 and 19.0 km, with an average
position of [17.2 ± 1.4] km, while the $LRT_{SHADOZ}$ varies between 14.0 and 16.0 km, with an average value of [14.9 ± 1.5] km, resulting in an average difference of [2.3 ± 0.3] between $CPT_{SHADOZ}$ and $LRT_{SHADOZ}$. The range of $CPT_{SHADOZ}$ values is in agreement with the results reported by Bègue et al. (2010), and with the average value obtained by Sivakumar et al. (2006) [17.2 ± (1σ) 0.6] km. On the other hand, the $LRT_{SHADOZ}$ range is below of Bègue et al. (2010) results, as well as, the average value is lower than that one reported by Sivakumar et al. (2006) [16.0 ± (1σ) 0.7] km. Consequently, the average difference
between $CPT_{SHADOZ}$ and $LRT_{SHADOZ}$ is higher than the values obtained by Bègue et al. (2010), Sivakumar et al. (2006), and Zhran et al. (2023) [0.91 ± (1σ) 0.15] km, [1.09 ± (1σ) 0.94] km, and 0.92 km, respectively. It is important to highlight that the number of $LRT_{SHADOZ}$ cases are lower than those of $CPT_{SHADOZ}$, as not all $T_{SHADOZ}(z)$ profiles have the essential characteristics for calculating the LRT.



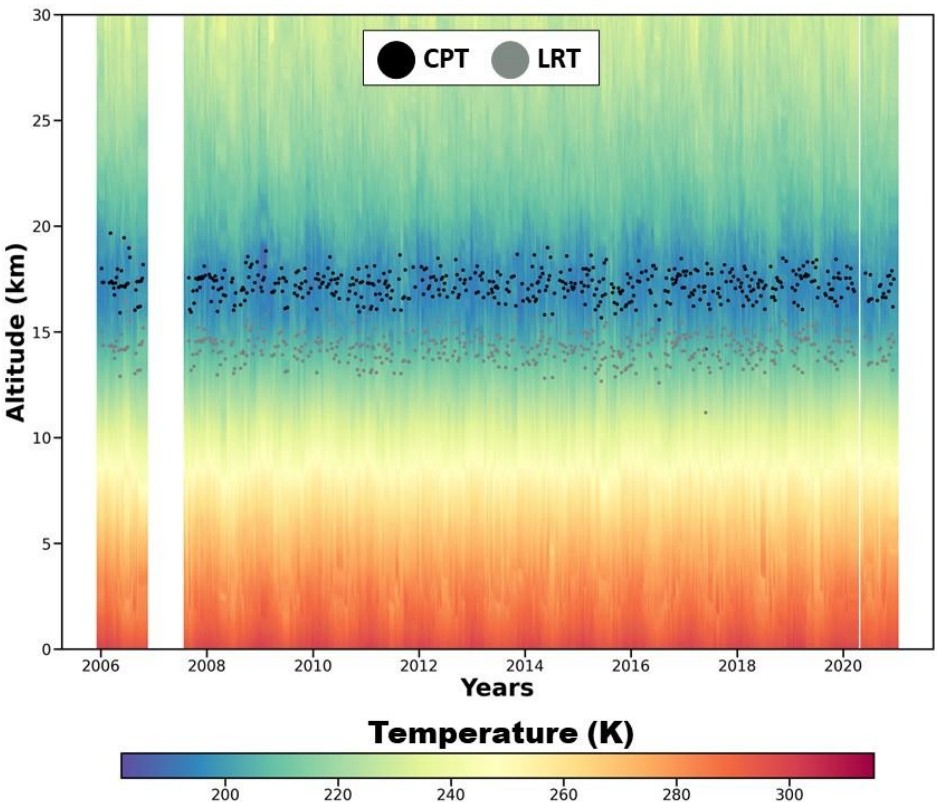

**Figure 3: Time-height temperature cross-section over Réunion from weekly balloon-sonde profiles from January 2006 to December 2020. Black and gray dots indicate the CPT and LRT heights, respectively.**

Similar to Figure 3, Figure 4 presents the curtain plot of the vertical temperature profiles from the surface to 60.0 km height, as derived from daily COSMIC-1 measurements from 2006 to 2020. By comparison to SHADOZ temperature profiles, COSMIC-1 profiles present a higher vertical range, up to the mesosphere (limited here to 60.0 km) with greater temporal sampling. Table 1 presents the total number of temperature profiles per month used in the present study from SHADOZ and COSMIC-1 measurements. COSMIC-1 presents a lack of data in February and March 2014, and from June to December 2019. In addition, based on the density of CPT and LRT points in Figure 4, it can be noted that there was a decrease in the number of COSMIC-1 overpasses over the study site from 2017.

A quick visual comparison of the $T_{SHADOZ}$ (Figure 3) and the $T_{COSMIC-1}$ (Figure 4) shows significant differences below 10 km altitude. The CPT values obtained from COSMIC-1 data ($CPT_{COSMIC-1}$) are located in the same range of $CPT_{SHADOZ}$, so that the average value provided by COSMIC-1 data [17.2 ± 1.3] km is similar to average $CPT_{SHADOZ}$ ([17.2 ± 1.4] km) and almost identical to value obtained by Sivakumar et al. (2006). Regarding the LRT ($LRT_{COSMIC-1}$, the range of values [between 14.0 and 16.0 km] and the average value ([14.7 ± 1.2] km) are similar to that one obtained from SHADOZ data, consequently the average difference between $CPT_{COSMIC-1}$ and $LRT_{COSMIC-1}$ [2.9 ± 1.9] km is similar to that one observed between the



SHADOZ data.  These results are in agreement with those presented by Xia et al (2021), who found an average difference of

2.67 km between $CPT_{COSMIC-1}$ and $LRT_{COSMIC-1}$.

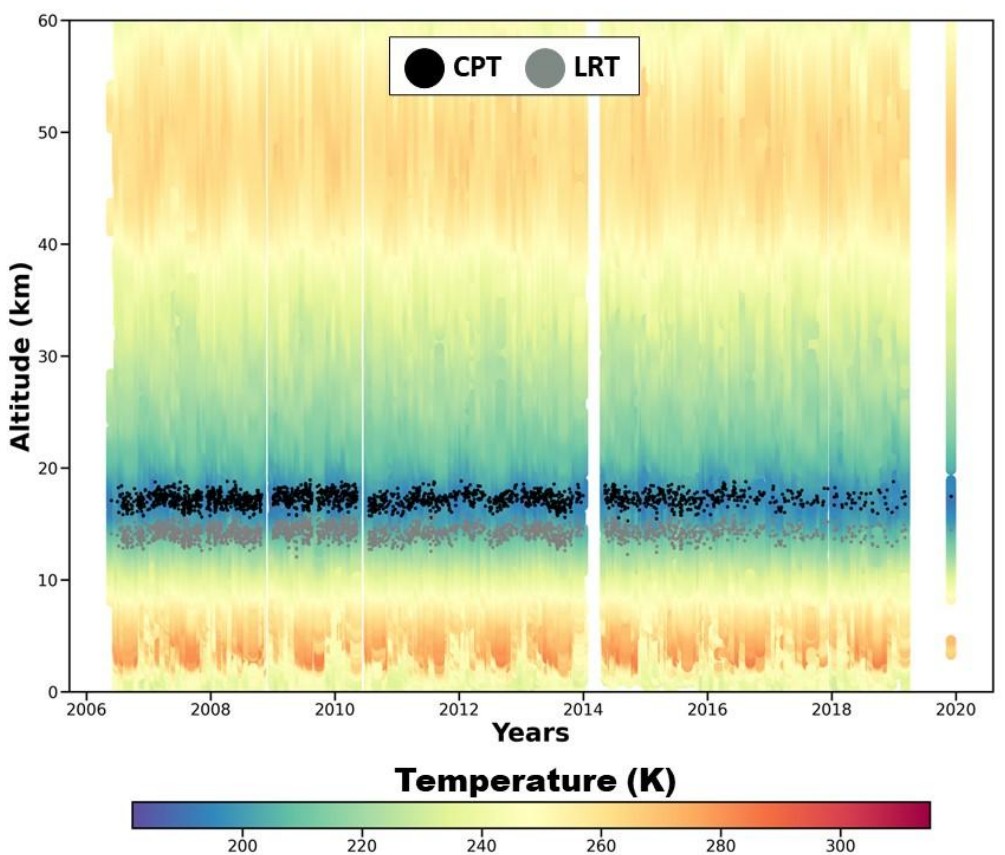

**Figure 4: Same as Figure 3 but concerning COSMIC-1 measurements over Réunion.**

**Table 1: Total monthly number of temperature profiles used from SHADOZ and COSMIC-1 measurements at the Réunion site.**

|          | JAN | FEB | MAR | APR | MAY | JUN | JUL | AUG | SEP | OCT | NOV | DEC | Total |
|----------|-----|-----|-----|-----|-----|-----|-----|-----|-----|-----|-----|-----|-------|
| SHADOZ   | 40  | 43  | 46  | 46  | 45  | 53  | 30  | 24  | 54  | 48  | 43  | 32  | 504   |
| COSMIC-1 | 159 | 163 | 197 | 175 | 158 | 169 | 223 | 216 | 162 | 164 | 112 | 99  | 1997  |

In contrast to SHADOZ and COSMIC-1 data, MERRA-2 does not show any data gap. Although the $T_{MERRA-2}$ have a

reasonable agreement with $T_{SHADOZ}$ and $T_{COSMIC-1}$ in tropopause region (Fig. 2), MERRA-2 data cannot be used to detect the

tropopause height accurately, because the heights are fixed, then the results obtained have low variability, making it impossible

to observe the variations and trends in this layer. Zou et al. (2023), when comparing CPT and LRT values using different

reanalysis databases (ERA-Interim, MERRA-2 and NCEP/NCAR Reanalysis 1), also address the difficulty of refinement due

to the coarser vertical resolution.





### 4. 3 Global Comparison

In this subsection, the global mean temperature profiles obtained from the 3 datasets (SHADOZ, COSMIC-1 and MERRA-2)
were computed and compared with each other. A global temperature profile is obtained by averaging all the temperature
profiles recorded by the same experiment over the entire study period (2006-2020). We thus obtained 3 global average profiles
for the 3 datasets: $\overline{T}_{SHADOZ}(z)$, $\overline{T}_{COSMIC}(z)$, and $\overline{T}_{MERRA}(z)$. They are superimposed in Figure 5(a), respectively in red, green
and blue dots. Figure 5(b) shows the temperature differences in the 0.0 - 30.0 km altitude range between the COSMIC-1 and
MERRA-2 global profiles and the SHADOZ global profile: $\left(\overline{T}_{SHADOZ} - \overline{T}_{COSMIC}\right)$ and $\left(\overline{T}_{SHADOZ} - \overline{T}_{MERRA}\right)$, respectively in
orange and black lines. Using SHADOZ data as a reference, it is evident that COSMIC-1 temperature values progressively
overestimate as height decreases below 8.0 km, whereas MERRA-2 assimilated temperatures show excellent agreement. The
largest differences are obtained between SHADOZ and COSMIC-1 values in the lower troposphere, up to 55.0 K nearby the
surface (see Figure 5b).

Above 10.0 km, the differences between $\overline{T}_{COSMIC}$ and $\overline{T}_{SHADOZ}$ reach approximately 1.0 K, continuing in this way until the
end of the $\overline{T}_{SHADOZ}(z)$ profile (around 30.0 km). $\overline{T}_{MERRA-2}(z)$ appears as a combination between $\overline{T}_{SHADOZ}(z)$ and
$\overline{T}_{COSMIC-1}(z)$, following the behavior of SHADOZ in the first 30.0 km, so that the average profiles do not present a difference
greater than 2.0 K, and follows COSMIC-1 in the rest of the profile. Such behavior was also observed by Tegtmeier et al.
(2020). The similarity of $T_{MERRA-2}$ with $T_{SHADOZ}$ and $T_{COSMIC-1}$ is justified by the composition of the MERRA-2 reanalysis
data, since in addition to being based on balloon-sonde profiles, these reanalysis data incorporate GPS-Radio Occultation
information, which did not happen with the original MERRA data.




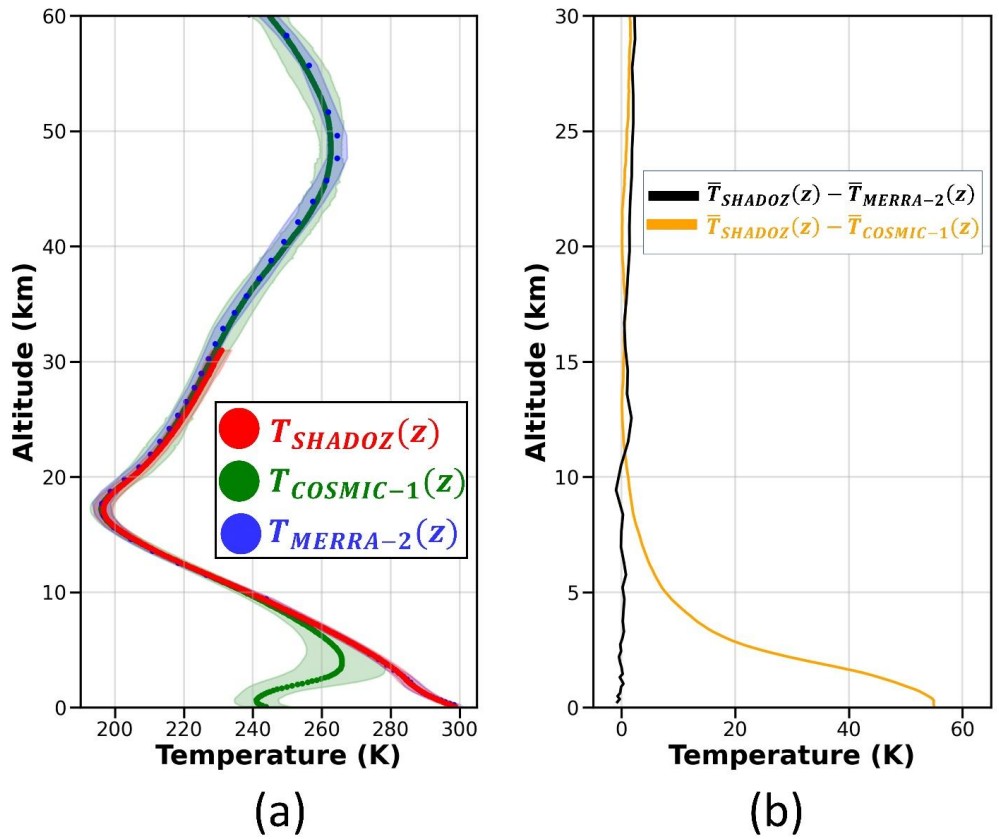

**Figure 5: (a) Global average temperature profiles over Réunion site from SHADOZ (red line), COSMIC-1 (green line) and MERRA-2 (blue line), framed with their ± 1σ (standard deviation) profiles (in coloured shadows). (b) Difference profiles: $\left(\overline{T}_{SHADOZ} - \overline{T}_{MERRA-2}\right)$ and $\left(\overline{T}_{SHADOZ} - \overline{T}_{COSMIC-1}\right)$, respectively in black and orange lines.**


### 4. 4 Seasonal Comparison

As reported by several authors (Seidel et al., 2001; Bencherif et al., 2006; Sivakumar et al., 2011; Bègue et al., 2010; Shangguan and Wang, 2022; Zhran and Mousa., 2023), the thermal structure of the atmosphere is seasonally dependent, notably in the tropics and subtropics. Thus, we examined and compared temperature profiles averaged by season (summer -DJF, autumn -

MAM, winter -JJA and spring -SON). Plots in the upper panel of Figure 6(a—d) superimpose the seasonal temperature profiles obtained from SHADOZ, COSMIC-1 and MERRA-2 and the associated seasonal temperature differences with the SHADOZ data as a reference (plots of the lower panel of Fig.6, e—h).

The altitude range of agreement between the seasonal $\overline{T}_{SHADOZ}$ and $\overline{T}_{COSMIC-1}$ temperature profiles vary depending on the season, as indicated in Figure 6(b). To enable comparison between different seasons, a horizontal dashed line is added to the

temperature difference profiles. This line shows the altitude at which the temperature difference between $\overline{T}_{SHADOZ}$ and




$\overline{T}_{COSMIC-1}$ begins to be lower than 5%. It appears from this line that the altitude of validity of the COSMIC measurements depends on the season. It is possible to identify a seasonal behavior, where the highest limit occurs in summer (5.3 km), then it decreases continuously, reaching 4.3 km in Autumn, and the lowest value in winter and spring (2.8 km). The best agreement between $\overline{T}_{SHADOZ}$ and $\overline{T}_{COSMIC-1}$ values are observed during the dry season (JJA), while during the wet season (DJF)
$\overline{T}_{COSMIC-1}$ profiles seem to underestimate the $\overline{T}_{SHADOZ}$ values over the widest range of altitudes. On the other hand, the difference between $\overline{T}_{SHADOZ}(z)$ and $\overline{T}_{MERRA-2}(z)$ does not present a seasonality. During all seasons, the differences do not exceed 3.0 K. The higher values are observed between 11.0 and 14.0 km, and above 19.0 km. In these regions difference between $\overline{T}_{SHADOZ}(z)$ and $\overline{T}_{MERRA-2}(z)$ is higher than that one observed between $\overline{T}_{SHADOZ}(z)$ and $\overline{T}_{COSMIC-1}(z)$.

The CPT values estimated by both databases are quite similar, so that when considering the uncertainty values, it can be stated
that the seasonal values obtained from COSMIC-1 and SHADOZ are practically coincident. The $CPT_{COSMIC-1}$ sometimes overestimates and at others underestimates the $CPT_{SHADOZ}$. The lower absolute difference between them is observed during the Winter (26 m), and the higher one in Autumn (122 m). The average $CPT_{SHADOZ}$ has a seasonal behavior similar to that one observed by Seidel et al. (2001), where the maximum and minimum values were observed in Summer and Winter, respectively. Bègue et al. (2010) also identified the maximum $CPT_{SHADOZ}$ during the summer, as well as Astudillo et al. (2020) using
Ground-Based GNSS Observations and Schmidt et al. (2004) using GPS RO data from the German CHAMP (CHAllenging Minisatellite Payload) satellite mission. This seasonal tropopause behavior also was observed by Sivakumar et al. (2011), which identified the minimum $CPT_{SHADOZ}$ during the winter, and Zhran and Mousa (2023). On the other hand, although the lowest values of $CPT_{COSMIC-1}$ occur during winter, in agreement with the results obtained by Seidel et al. (2001) and Sivakumar et al. (2011), the maximum values occur in autumn, what is not in agreement with the current literature. It is
important to highlight that this seasonal behavior reinforces the influence of solar radiation in tropopause height, mainly in tropical and subtropical regions (Sivakumar et al., 2011).

Regarding the $CPT_{SHADOZ}$ temperature, the average monthly values obtained are similar, although a little bit smaller than those estimated by Bègue et al. (2010), so that the lower and higher mean absolute difference are -0.1 K (February, the coldest month identified by Bègue et al. (2010)) and -1.5 K (September, the hottest month identified by Bègue et al. (2010)), respectively. In
this work the coldest month is January ([193.1 ± 4.6] K) and the hottest is October ([197.1 ± 4.6] K). Considering $CPT_{COSMIC-1}$, excepting May, where the average $CPT_{COSMIC-1}$ is 2.7 K higher than the $CPT_{SHADOZ}$ obtained by Bègue et al. (2010), all other months have a smaller average temperature value, so that the lower and higher absolute difference are -0.3 (August) and -2.7 (April), respectively. The coldest and the hottest month are February ([192.1 ± 4.6] K) and September ([197.5 ± 4.8] K), as observed by Bègue et al. (2010), respectively.



**Figure 6:** In the upper part is presented a seasonal comparison ((a) DJF, (b) MAM, (c) JJA, and (d) SON) among $T_{SHADOZ}(z)$ (red), $T_{COSMIC-1}(z)$ (green) and $T_{MERRA-2}(z)$ (blue) profiles, and their respective standard-deviations. In the lower part is presented a seasonal comparison ((e) DJF, (f) MAM, (g) JJA, and (h) SON) among the differences of $T_{SHADOZ}(z)$ and $T_{MERRA-2}(z)$ (black line) and $T_{SHADOZ}(z)$ and $T_{COSMIC-1}(z)$ (orange line). The dotted purple line represents the height where the difference between $T_{SHADOZ}(z)$ and $T_{MERRA-2}(z)$ is lower than 5%.





Due to the necessity of data above 40 km, only $T_{COSMIC-1}$ was applied to estimate the stratopause height, as indicated previously in section 3.3. The stratopause appears to be highest in spring ([48.9 ± 4.6] km) with a maximum temperature of [266.3 ± 0.4] K, whereas it is lowest in summer ([48.5 ± 4.2] km) with a temperature maximum of around [267.2 ± 0.8] K.

These results are in agreement with the outcomes of other studies conducted at tropical locations. Batista et al. (2009) analyzed 14 years (from 1993 to 2006) of temperature profiles recorded by a sodium resonance LiDAR at São José dos Campos, Brazil (23°S, 46°W). They found that the local stratopause altitude was ~49km, with the maximum temperature varying from 265 to 270K. Moreover, Sivakumar et al. (2011b) used temperature profiles over Réunion recorded between 1994 and 2007 by a Rayleigh LiDAR and reported that the stratopause height occurrence was in the 47–49 km height range, with temperatures

ranging from 260 K to 270 K. In addition, the same seasonal behavior of the stratopause height presented in Figure 6 was observed by France et al. (2012) and Vignon et al. (2015) in their climatology studies using the Microwave Limb Sounder (MLS) and reanalysis data (MERRA-2).

## 4. 5 Combination between SHADOZ and COSMIC-1 datasets

Considering the results presented in the previous section, where SHADOZ and COSMIC-1 temperature profiles showed a

good agreement in the 10-28 km altitude range, we merged the two datasets to construct quasi-continuous and regular space-by-time matrix temperature values. Whenever it was impossible to measure temperature profiles by balloon-sonde due to technical, meteorological or logistical convenience, we filled the gaps with COSMIC-1 profiles. Then, the daily temperature series obtained were reduced to monthly and kilometric averages and are presented in Figure 7. This new dataset will be applied in the trend analyses (section 5).

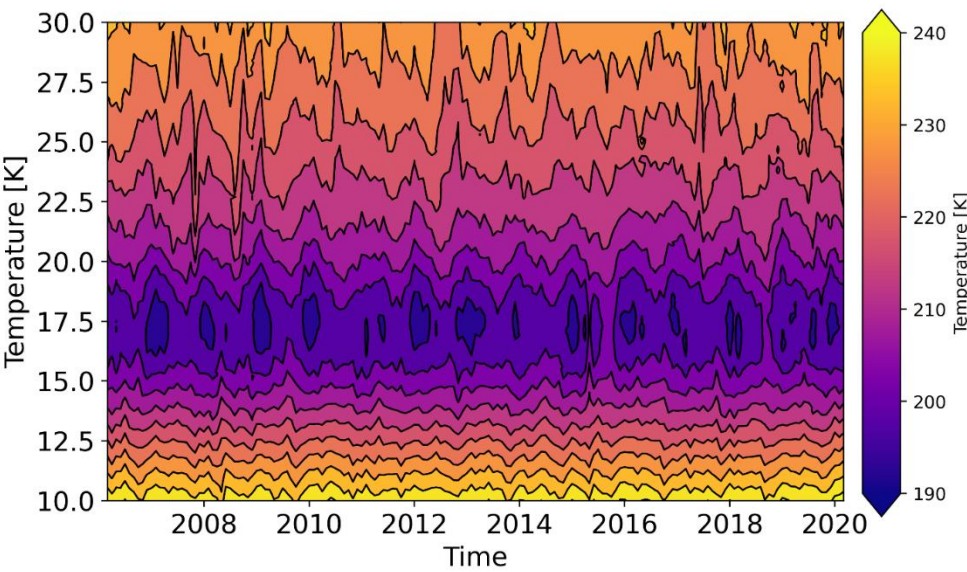


**Figure 7: Monthly time-height temperature cross-section over Réunion constructed by combining SHADOZ and COSMIC-1 profiles.**





## 5 Trend Model Analysis

For trend analysis, we used the multiple linear regression method taking into account the same *forcings* used by Bègue et al.
(2010) and by Toihir et al. (2018): annual and semi-annual cycles, quasi-biennial oscillation, and ENSO (section 3.4).

The analysis of the $R^2$ profile, as depicted in Fig.8a, reveals two layers where the Trend-Run model performs well and explains more than 65% of the temperature variability: one layer at around 10 km and another at around 18 km. Additionally, the $R^2$ profile shows two minima (~40%), where the Trend-Run model performs less well, in the upper troposphere (14 km) and the upper stratosphere (26–27 km). This result is not surprising given the short length of the time series (2006–2020). We then
examine temperature trends in two tropospheric heights (10 km and 15 km) and two stratospheric layers (19 km and 24 km), in addition to temperature trends at the local tropopause. Figure 8b overlays the profiles of the considered *forcings* as derived from the Trend-Run model. Overall, the annual oscillation (AO) emerges as the most dominant forcing, particularly in the lower stratosphere, where it accounts for over 40% of the variability at approximately 19 km. The semi-annual oscillation (SAO) exhibits its maximum contribution in the troposphere, specifically at 10–11 km. The ENSO forcing displays an absolute
maximum contribution of -20 % in the lower stratosphere (22–23 km) and has a nearly constant contribution of approximately 10% in the troposphere. The QBO index, obtained for the 70-hPa pressure level, shows almost no contribution in the troposphere and a quasi-constant contribution in the stratosphere (~4%).

Following Figure 8(a), the higher determination coefficient can be observed in the UTLS region (18 – 20 km), so the higher contribution, in this region, in the trend model is from AO (Fig. 8b). In addition, such a region presents an increase of around
0.25 K/decade (Fig. 8c).

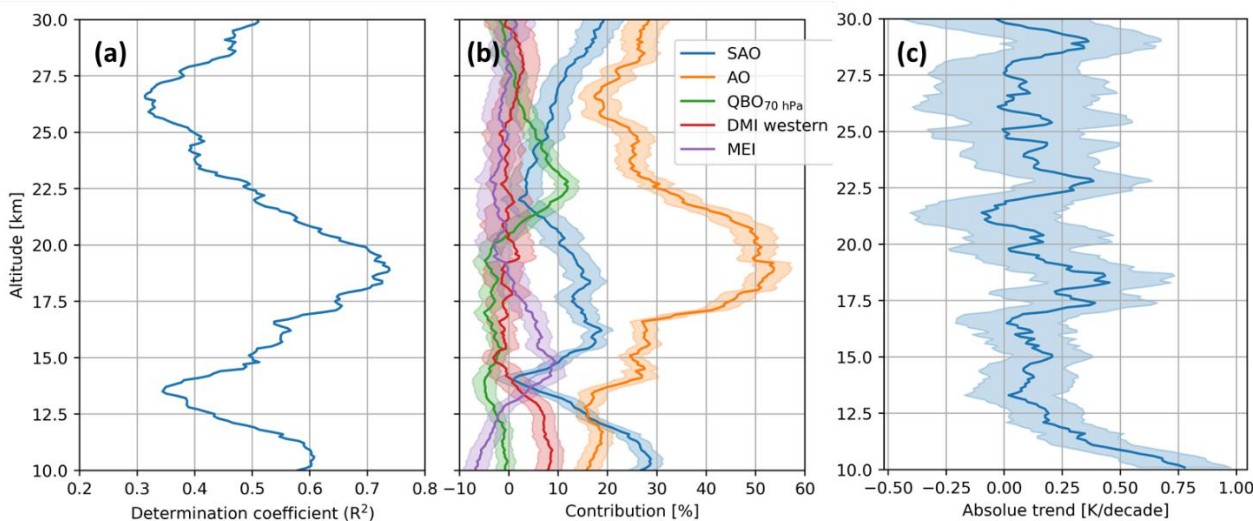

**Figure 8: (a) autocorrelation coefficient profile, (b) vertical profile of contribution of each forcing, (c) vertical profile of absolute variation of temperature per decade.**



Figure 9 shows temperature trends at the tropopause, CPT (Fig.9a) and LRT (Fig.9b). Temperature at the CPT presents a significant decreasing trend of [-0.13 ± 0.25] K/decade, where seasonal cycles (AO, SAO) seem to be the most dominant forcing. Zou et al. (2023), using ERA5 data, observed a tropopause cooling of [-0.09 ± 0.03] K/decade from 1980 to 2021. Although the time interval (1979-2005) is different from the present study, Tegtmeier et al. (2020) reported a cooling of the tropical CPT from -0.3 to -0.6 K/decade.

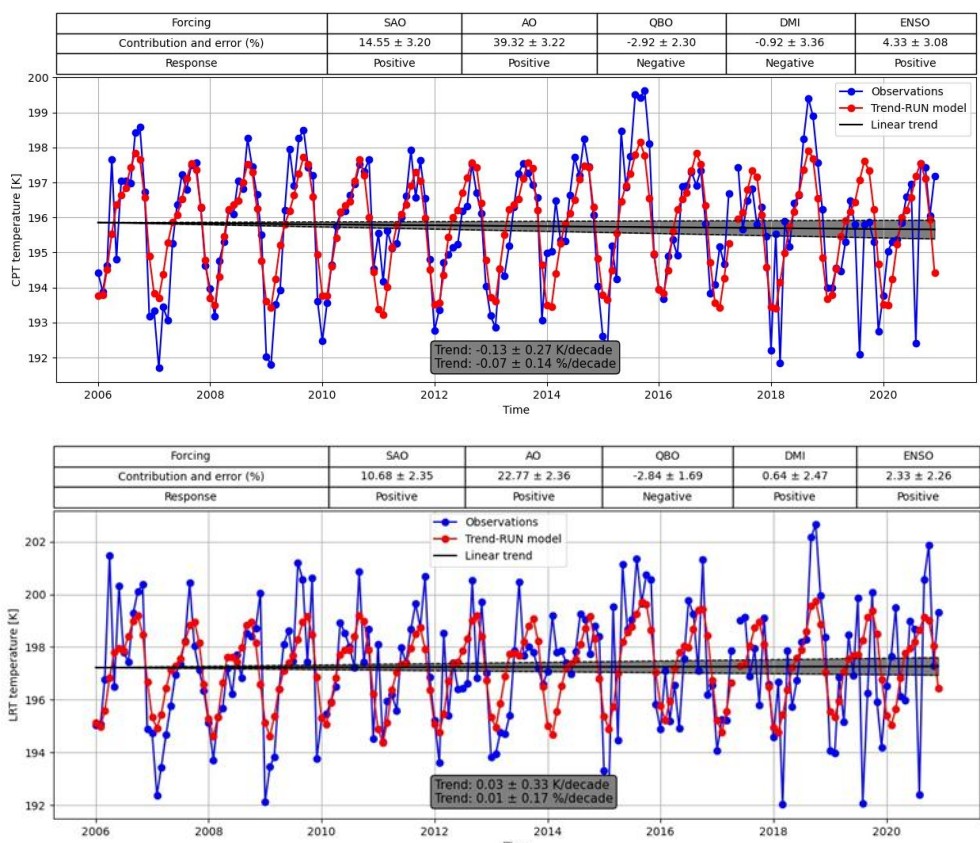

**Figure 9: Trend model of CPT (a) and LRT (b) temperature variation.**

On the other hand, LRT temperature (Fig. 10) shows a little non-significant increase ([0.03 ± 0.33] K/decade), where in the same way as CPT temperature, the seasonal cycles (AO, SAO) are the most dominant forcing. Shangguan and Wang (2022) and Negash and Raju (2024) also observed a strong influence of AO and SAO in the UT-LS region from COSMIC-1 and ERA-5 data, and COSMIC-1 and Radiosonde data respectively in subtropical latitudes. Following Frierson (2006), the latent heat release caused by thermal forcing in the troposphere is what creates the strong AO in temperature. In addition, according to van Loon and Jene (1969), tropical and sub-tropical SAO are most pronounced in the areas where the Intertropical Convergence Zone crosses the Equator twice a year, in particular, from Eastern Africa to the central Pacific Ocean, which coincides with the region where Réunion is located.





## 6 Conclusions

Tropopause temperature and height serve as key indicators of anthropogenic climate change, influenced by factors such as
stratospheric ozone, greenhouse gas concentrations, and volcanic activity. However, monitoring their variability remains
challenging due to the sparse distribution of observation stations, particularly in the Southern Hemisphere. To address this, we
compared temperature profiles from three datasets—SHADOZ, COSMIC-1, and MERRA-2—to assess their similarities and
differences and to develop a refined dataset for trend analysis. Our analysis of SHADOZ and COSMIC-1 data (2006–2020)
revealed strong agreement in temperature profiles above 10 km. MERRA-2 data showed a good correlation with SHADOZ up
to 30 km and with COSMIC-1 above 30 km, but its coarse vertical resolution limited its applicability for tropopause height
estimation. Using the Cold Point Tropopause (CPT) and Lapse Rate Tropopause (LRT) methods, we found that CPT-derived
tropopause heights and temperatures were consistent across SHADOZ and COSMIC-1, whereas LRT values varied more due
to differences in vertical resolution. Comparisons within each dataset confirmed that LRT-derived tropopause heights were
systematically lower than those from CPT, while LRT temperatures were higher—consistent with previous studies.
Additionally, CPT exhibited seasonal variability, with higher values in summer and lower values in winter. To enhance data
coverage, we created a new dataset by integrating COSMIC-1 data into SHADOZ profiles between 10 and 30 km, preserving
the seasonal characteristics observed in both datasets. This combined dataset improves the representation of tropopause
dynamics in regions with sparse observations. Trend analysis highlighted the significant influence of the annual oscillation
(AO), particularly in the upper troposphere–lower stratosphere (UT-LS) region. We observed a decreasing trend in CPT
temperature (-0.13 ± 0.25 K/decade) and a slightly increasing trend in LRT temperature (0.03 ± 0.33 K/decade), both
predominantly influenced by AO and the semi-annual oscillation (SAO). Our findings demonstrate the value of COSMIC-1
data in studying tropopause dynamics, enabling the extension of time series in regions with radiosonde observations and
providing critical data where in situ measurements are unavailable. This study underscores the potential of satellite-based
remote sensing to enhance our understanding of climate-related changes in the tropopause.

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
