# Peer review of "Variability and trend analysis of temperature in the upper troposphere and stratosphere region over the tropics (Réunion), by combining balloon-sonde and satellite measurements"

_EGUsphere, 2025_

## Referee Comment (RC3)

Review report of the paper "Variability and trend analysis of temperature in the upper troposphere and stratosphere region over the tropics (Réunion), by combining balloon-sonde and satellite measurements" by Gregori de Arruda Moreira et al.

The paper "Variability and trend analysis of temperature in the upper troposphere and stratosphere region over the tropics (Réunion), by combining balloon-sonde and satellite measurements", authors investigated how temperature and the height of the tropopause are changing over the tropics near Réunion Island using 15-year data from radiosonde, COSMIC-1 and MERRA-2. Using the combined datasets, they assess how well different methods match when estimating the tropopause height to explore a refined long-term trend. Studying tropopause temperature and height variability is very important for understanding the anthropogenic contribution to climate change, which is a highly relevant topic in ANGO. Given the sparse availability of datasets from the Southern Hemisphere, a long-term report on this subject from Réunion is particularly valuable and much needed. However, the referee feels that the paper currently lacks some important details, and the authors should provide further clarification on following concerns before the manuscript can be considered ready for acceptance.

**Major concerns:**

**1. Methodology:**

The methodology section requires more detail and clarification. The authors state that they used radiosonde temperature profiles from 2006–2020 (available weekly) together with COSMIC-1 RO data within a margin of  $\pm 2^{\circ}$  and  $\pm 3^{\circ}$  spatial resolution. However, it is not clearly described how simultaneous these datasets were. Although they showed examples of simultaneous measurements in Figure 2, a clear description should be given in the methodology. As in Table 1 indicates a larger number of COSMIC profiles, but it is not specified whether the authors computed weekly averages of these data to match the radiosonde records, or how day-to-day variability was accounted for. Although a long-term analysis, a clear description of data is required for the reliability of the derived tropopause characteristics.

Another concern about the Trend-Run model, the authors used a simplified approach to evaluate trend by only reporting the coefficient of determination (R2). However, R2 alone cannot provide information about the reliability or statistical significance. The authors are encouraged to include uncertainty analyses, for example confidence intervals and performing sensitivity tests to demonstrate which forcings have the most significant influence on the results.

**2. Lack of proper discussion:**

Using the combined dataset, the authors identified differences/similarities in estimating tropospheric heights both CPT and LRT and reported some seasonal variations and small but significant long-term changes in tropopause temperature and height, which are important for understanding climate change. However, they did not properly discuss these results in the broader context of atmospheric dynamics, radiative forcing, or anthropogenic climate forcing. The

discussion is limited to a brief comparison with Sivakumar et al. (2011) and Bègue et al. (2010) etc. The authors should expand the discussion of the results with additional published works and explain how their results align or diverge from global findings, focusing on the uniqueness or similarities of TTL dynamics in the southern hemisphere. Without a clear discussion and interpretation of the results, the manuscript remains incomplete in its current form.

**Minor concerns:**

- 1. The introduction could be improved by more clearly highlighting key gaps in studying tropopause height and temperature over the tropics, and particularly importance of study over Réunion. While this section is otherwise well written and easy to follow, motivation of the study would be stronger if the authors emphasized these gaps more explicitly.
- 2. Line 100: I feel the sentence 'Therefore, temperature gradients ..... and chemistry.' is irrelevant here.

---

## Author Comment (AC1)

**DETAILED ANSWER TO REVIEWER #1**

First, we would like to thank Reviewer #1 for all the comments, queries, and suggestions for modification. We considered all of them and tried to answer all queries. We transcribed the reviewer comments below, organizing them in a sequential order to help their identification in the revised version of the manuscript.

The present manuscript deals with temperature and tropopause height measured over a tropical site, Réunion (21.10° S; 55.48° E), an island in Indic Ocean from 2006 to 2020. Three types of measurements are used to provide data: local balloon-borne temperature profiles (SHADOZ), satellite by using GNSS-RO technique at COSMIC-1, and reanalysis data from MERRA-2. Tropopause heights are obtained using the Cold Point Temperature (CPT) and/or Lapse Rate Temperature (LRT) methods. Those measurements are compared and show a good consistency, mainly from 10 to 30 km height between SHADOZ and COSMIC-1. So, their data was combined for constructing a more refined dataset, which was then used to fit a model which considers the AO, SAO, QBO, ENSO, SSN and IOD, as the linear trend. As expected, the analysis indicated a main influence of AO and SAO oscillations in Tropopause height dynamics, as well as a decreasing trend in CPT and a slight increase in the Lapse Rate Tropopause (LRT) height. The paper is well written and well-structured, the methodology seems to be correct, and the subject has merits and is of interest for Annales Geophysicae. However, this reviewer has concerns about some points that should be addressed by the authors before an acceptance recommendation. The authors should provide a complete revision of the references as indicated below. Therefore, as the subject of the paper is very interesting, I am ready to recommend this manuscript for publication after taken into consideration the above points and after giving a detailed response to my comments.

**Minor comments:**

- Line 31: "Niño" not "Ninõ"
*Done*

- Line 334:  Fig. 10 is Fig. 9 and the plots should contain a) and b).
*Done*

Also, in relation to Figure 9 what are the meaning of the two slopes inside the gray rectangles?

*The solid black lines represent the CPT and LRT temperature trends in figures a and b, respectively. The dotted black lines, in both figures, indicate the limits of the standard deviation of trends. As a general note, trends are usually noted at the 2 sigma level. Only trends whose value is greater than the 2 sigma threshold are caraterized as 'significant'. All trends in this paper are indicated with standard deviation at the 1 sigma level. In order to improve the main document, the caption of figure 9 has been changed to:*

*"Figure 9: Trend model of CPT (a) and LRT (b) temperature variation. The black lines represent the trend of CPT and LRT temperature, in figures a and b, respectively. The dotted black lines, in both figures, represent the standard deviation of the temperature trend at the 1 sigma level."*

Additionally, do the slopes lines have deviation increasing with time? Justify or correct.

Deviation increases with time because this is how trend uncertainty is computed in the model. The standard deviation associated with the trend value corresponds to uncertainty in the trend slope. It is important to consider that two successive measurements are not necessarily independent. The degree of dependence between successive measurements is evaluated using the autocorrelation coefficient. This coefficient makes it possible to determine the uncertainty on the estimated trend and on the contribution of each forcing. The trend uncertainty, therefore, represents the uncertainty in the trend slope value. Therefore, in order to improve the main text, the following text has been added:

Line 326
*"Figure 10 shows temperature trends at the tropopause, CPT (Fig.10a) and LRT (Fig.10b), as well as, their standard deviations at the $1\sigma$ level (dotted line). So that the standard deviation associated with the trend value corresponds to uncertainty in the trend slope. Temperature at the CPT presents a significant decreasing trend of [-0.13 ± 0.25] K/decade, where seasonal cycles (AO, SAO) seem to be the most dominant forcing"*

**References:**

There are many inconsistencies in the references with works cited but not listed and listed and not cited. Although this is not a reviewer's duty I have patiently indicated to the authors the required changes as follows.

1. Austin, J., and T. J. Reichler, Long-term evolution of the cold point tropical tropopause: Simulation results and attribution analysis, J. Geophys. . Res, 113, D00B10, 2008. It appears in the references list, but it is not mentioned in the text. Done
2. Astudillo et al., 2014, Astudillo et al., 2020 and Anthes et al., 2008 are cited in text but are not in the References list. Done
3. Birner et al., 2006 is cited in text but is not in the References list.
4. Cheng et al, 2006 is cited in text but is not in the References list.
5. Dameris et al., 1995 is cited in text but is not in the References list.
6. Fueglistaler et al., 2009 is cited in text but is not in the References list. Done
7. Hoinka, 1998 is cited in text but is not in the References list. Added
8. Li et al., 2008 is cited in text but is not in the References list.
9. Ladstädter, F., Steiner, A. K., and Gleisner, H. Resolving the 21st century temperature trends of the upper troposphere–lower stratosphere with satellite observations. Sci. Rep. 13, 1306, 2023. It appears in the references list, but it is not mentioned in the text.
10. Morioka et al., 2010 is cited in text but is not in the References list.
11. Mateus, P., Mendes, V. B., and Pires, C. A. Global Empirical Models for Tropopause Height Determination. Remote Sens., 14, 4303, 2022. It appears in the References list, but it is not mentioned in the text.

12. Randel and Cobb, 1994, Randel et al., 2000 ,Randel and Jansen, 2013, Reid and Gage, 1981, Reid and Gage, 1984 and Reid and Gage, 1985 are cited in text but are not in the References list.

13. Sivakumar et al., 2006 and Sivakumar et al., 2017 are cited in text but are not in the References list.

14. There are two references Sivakumar et al., 2011. Check what is a) and b).

15. Santer et al., 2004 is cited in text but is not on the References list.

16. Saji et al., 1999 is cited in text but is not in the References list.

17. Selkrik, 1993 is cited in text but is not in the References list.

18. Sterling, C. W., Johnson, B. J., Oltmans, S. J., Smit, H. G. J., Jordan, A. F, Cullis, P. D., Hall, E. G., 400 Thompson, A. M., Witte,J. C. Homogenizing and Estimating the Uncertainty in NOAA's Long Term Vertical Ozone Profile Records Measured with the Electrochemical Concentration Cell Ozonesonde, Atmos. Meas. Tech. 11, 3661-3687, 2018. It appears in the References list, but it is not mentioned in the text.

19. Thompson, A. M., Witte, J. C., Sterling, C., Jordan, A., Johnson, B. J., Oltmans, S. J., and Thiongo, K. First reprocessing of Southern Hemisphere Additional Ozonesondes (SHADOZ) ozone profiles (1998-2016): 2. Comparisons with satellites and ground-based instruments. Journal of Geophysical Research: Atmospheres, 122, 13,000-13,025, 2017.

It appears in the References list, but it is not mentioned in the text.

20. Xian, T. and H.omeyer, C. R. Global tropopause altitudes in radiosondes and reanalyses, Atmos. Chem. Phys., 19, 5661–5678, 2019. It appears in the References list, but it is not mentioned in the text.

21. Wang , J. S., Seidel, D. J., and Free, M. How well do we know recent cl imate trends at the tropical tropopause? J. Geophys. Res. Atmos. 117, D09118, 2012.

Weyland, F., Hoor, P., Kunkel, D., Birner, T., Plöger, F., and Turhal, K.: Long-term changes in the thermodynamic structure of the lowermost stratosphere inferred from reanalysis data, Atmos. Chem. Phys., 25, 1227–1252, https://doi.org/10.5194/acp-25-1227-2025, 2025.

Witte, J. C., Thompson, A. M., Smit, H. G. J., Fujiwara, M., Posny, F., Coetzee, G. J. R., Northam, E. T., Johnson, B. J., Sterling, C. W., Mohamad, M., Ogino, Shin-Ya, Jordan, A., and da Silva, F. First reprocessing of Southern Hemisphere ADditional OZonesondes (SHADOZ) profile records (1998-2015): 1. Methodology and evaluation, J. Geophys. Res. Atmos., 122, 6611-6636, 2017.

Witte, J. C., Thompson, A. M., Smit, H. G. J., Vömel, H., Posny, F., and Stübi, R. First reprocessing of Southern Hemisphere ADditional OZonesondes profile records: 3. Uncertainty in ozone profile and total column. Journal of Geophysical Research: Atmospheres, 123, 430 3243-3268, 2018.Appear in the References list but are not mentioned in the text.

22. WMO,1957 is cited in text but is not in the References list.

*We apologize for this mistake. All references have been properly cited in the main text or removed as indicated in the marked version of the new document*

---

## Author Comment (AC2)

**DETAILED ANSWER TO REVIEWER #2**

First, we would like to thank Reviewer #2 for all the comments, queries, and suggestions for modification. We considered all of them and tried to answer all queries. We transcribed the reviewer comments below, organizing them in a sequential order to help their identification in the revised version of the manuscript.

The authors presented a manuscript entitled, "Variability and trend analysis of temperature in the upper troposphere and stratosphere region over the tropics (Réunion), by combining balloon-sonde and satellite measurements". Integration of datasets from various instruments, such as SHADOW, COSMIC-1, and MEERA-2, was used in the manuscript to assess the temperature in the upper Troposphere and Stratosphere region. The results are consistent. However, I identified some moderate-to-major issues that I believe should be properly addressed before the manuscript can be considered for publication. Below, I provide some major comments, along with minor corrections.

**Major comments:**

1 - The manuscript title emphasizes more on troposphere temperature rather than height; however, in the abstract part more emphasis is given to height relatively which break the flow/continuity (as per title). Hence, either the abstract can be modified accordingly, giving more emphasis on temperature, or the title can be modified (such as "Variability and trend analysis of temperature and height in the upper troposphere and stratosphere region over the tropics (Réunion), by combining balloon-sonde and satellite measurements").

We thank the referee and agree with their comment and suggestion. The title has been changed to:

*"Variability and trend analysis of temperature and height in the upper troposphere and stratosphere region over the tropics (Réunion), by combining balloon-sonde and satellite measurements"*

In addition, the abstract has been changed:

Line 12:

*"This study compares tropopause height and temperature estimates from in-situ and remote …"*

2 - In the introduction part, the authors has to discuss, the possible gaps in the previous study (if any) and how this study fill the gap/improves the existing results or different from the previous study.

We thank the reviewer for this comment. The main improvement of our study is to demonstrate the feasibility of combining different data sources to create a more detailed timeseries of the Tropopause characteristics. In order to clarify this point in the main text, the following text has been added:

Line 40:

*"Considering remote sensing, Global Navigational Satellite System Radio Occultation (GNSS-RO) stands out for offering accurate tropospheric profiles with high vertical resolution and global coverage independently of weather conditions, however, are endowed of lower temporal resolution. Therefore, considering the limitations and advantages of each methodology, an option to improve the tropopause monitoring is to combine them. Although, firstly it is necessary to identify their similarities and differences.*

*In this context, this study compares vertical temperature profiles from the Southern Additional Ozonesondes (SHADOZ) network, Constellation Observing System for Meteorology Ionosphere and Climate 1 (COSMIC-1), and Modern Era Retrospective analysis for Research and Applications – Version 2 (MERRA-2) over Réunion (2006–2020) to identify similarities and/or differences."*

3 - In the methods section, three methods (radiosonde, COSMIC-1, MEERA-2) are discussed to measure the tropopause temperature and height.

3.1 What are their limitation with respect to each other and the variation in the obtained results?

- The radiosonde has limited spatial and temporal resolution. Such a characteristic can hinder a detailed observation of the dynamics of the tropopause.
- COSMIC-1 has as its main limitation the data gaps during maintenance periods, and progressive degradation that began in 2019 resulting in inoperability starting in May 2020.
- MERRA-2 has as its main limitation the fixed height values in the temperature profile. Such a characteristic can make it impossible to observe variations and trends in the tropopause behavior.

    To clarify these points, the following phrases have been added in the main text:

Line 72

*"The radiosonde technique has as its main disadvantages the limited horizontal and temporal resolution, which can hinder a detailed observation of the dynamics of the tropopause."*

Line 83

*"Its main limitation is the data gap during maintenance periods and after the beginning of the progressive degradation, which began in 2019 and resulted in total inoperability in May 2020."*

Line 97

*"These fixed heights can make it impossible to adequately observe some variations and trends in the tropopause behavior."*

3.2 For analysis of temperature profile (Figure 2d,e,f), the authors are taking Tshadow(Z) as base and calculating the TSHADOW(Z) - TCOSMIC-1(Z) and Tshadow(Z) - TMEERA-2(Z) . Is there any specific reason for it?

Yes. The radiosonde (SHADOW) is the reference instrument because COSMIC-1, like other satellites, was validated and calibrated from ground-based data and has limitations in its vertical profile, mainly in the lower troposphere, as demonstrated in section 4.1. MERRA-2, as mentioned in section 4.3, is a reanalysis model that incorporates balloon-sonde profiles and GPS-Radio Occultation information. Therefore, MERRA-2 temperature profiles can be a combination of COSMIC-1 and SHADOZ data, and consequently, it cannot be a reference instrument.

3.3 What happens if we take either TCOSMIC-1(Z) or TMEERA-2(Z) as base?

As indicated in the previous answer, the COSMIC-1 data presents problems in the first five kilometers of the troposphere; therefore, if used as a reference, it would result in comparisons that could incorrectly underestimate or overestimate the other equipment. Regarding MERRA-2, being a composite of the other two systems, its use as a reference is unfeasible, but if used, it would have excellent agreement with the radiosonde in the lower part of the profile and excellent agreement with COSMIC-1 in the upper part of the profile, as shown in Figure 5a.

4 - In method section, Trend-Run linear regression model used. Is there any statistical model (linear/non-linear) that can also be used for the available dataset?

We believe that other techniques can be applied. However, trend-run linear regression is a classic Multi-Linear Regression (MLR) model, which is the most widely used technique in this type of analysis. Therefore, in order to conduct a discussion with the main works cited in the literature, trend-run linear regression was selected.

5 - Section 4.2 Weekly and daily profiles: Is it weekly and daily profiles or weekly and monthly profiles? Please check

In this section, we present the Weekly temperature profiles of SHADOZ, and the daily temperature profiles of COSMIC-1. In order to clarify this point, the section title has been changed to:

Line 178

*"4.2 Weekly (SHADOZ) and daily (COSMIC-1) temperature profiles"*

6 - Under section 4.2: In the line no. 207, it is mention that "In contrast to SHADOZ and COSMIC-1 data, MERRA-2 does not show any data gap". However, the Time-height temperature cross-section by MEERA-2 is not shown (such as Figure 3 and Figure 4). Please add a Time-height temperature plot corresponding to MEERA-2, also (if possible) for better clarity and visualization of readers.

This new figure was added:

[Figure]

Figure 5: Same as Figure 3 and 4 but concerning COSMIC-1 measurements over Réunion.

7 - In the seasonal comparison section, it is mentioned that the thermal structure of the atmosphere is seasonally dependent, notably in the tropics and subtropics (line no. 238). With reference to the present study, are the obtained results valid in the region present only in the tropics and subtropics region around the globe? and how to check the robustness of the obtained results (if any)?

As described in section 4.4, this statement is based on the data and assertions presented in the following works:

Seidel et al., 2001; Bencherif et al., 2006; Sivakumar et al., 2011a; Bègue et al., 2010; Shangguan and Wang, 2022; Zhran and Mousa., 2023

The results obtained in this work reinforce this assertion. In addition, section 4.4 presents a comparison between the results obtained in this work and the results discussed in the references previously indicated. Therefore, this demonstrates the robustness of the results obtained.

**Minor:**

1- In the section 2 (under materials), experiments no. may be marked as 2.2a, 2.2b, 2.2c, respectively, as they are all different experiment. Or they can be mention in a single section separated with subsections.

Done

2 - Sivakumar (2011), and Sivakumar (2011b, line no. 288) are mentioned in the citation but not marked properly in the references.

Done

3 - Figure 1: mention this figure in the main text body. In the caption, please mention what the blue balloon symbol represents. In case if it represents the study area, the location of this blue balloon in the map and the provided latitude/longitude are mismatched. Kindly recheck and correct it. Also, mark the Roland Garros International Airport on the map. The latitude/longitude of the study site, Airport, and map should be in uniform (either in Decimal Degree or in Degree Minute Second). Mark the location of the map in the inset map by an arrow or a square box.

Done

[Figure]

4 - Figure 7 & 9: mark time in years as Time (years).

Done

5 - Line 144: "In this section is performed…….." may be rearranged as, "In this section, a comparison is performed among……"

Done

6 - Figure 2 (d, e, f): In the caption, it mentions that the difference between Tshadow(Z) and TCOSMIC-1(Z) is represented by black line, while in the legend it is marked as orange color. Similarly, with the Tshadow(Z) and TMEERA-2(Z). Please check and correct it.

We apologize for this mistake.

The caption was rewritten:

Line 162

"Figure 2: Comparison between $T_{SHADOZ}(z)$ (red), $T_{COSMIC-1}(z)$ (green) and $T_{MERRA}$ (z) (blue) profiles on 25-06-2014 (a), 19-11-2014 (b) and 17-09-2014 (c) and the difference between $T_{SHADOZ}(z)$ and $T_{COSMI}$ (z) (orangeblack line) and $T_{SHADOZ}(z)$ and $T_{MERRA}$ (z) (blackorange line) profiles to the same days (d), (e), and (f), respectively."

**Citation & References:**

Most of the citations provided in the main text body are missed in the reference and vice-versa (such as in the Introduction section, the citations- Fueglistaler et al., 2009; Randel and Jensen, 2013; Astudillo et al., 2014; Santer et al., 2004; Reid and Gage, 1981, 1984, 1985; Randel et al., 2000) are missed in the reference part. Similarly, in the reference part, almost 33% references are missed/not cited in the main text body. Please check them and insert/remove accordingly.

We apologize for this mistake. All references have been properly cited in the main text or removed as indicated in the marked version of the new document

---

## Author Comment (AC3)

**DETAILED ANSWER TO REVIEWER #3**

First, we would like to thank Reviewer #3 for all the comments, queries, and suggestions for modification. We considered all of them and tried to answer all queries. We transcribed the reviewer comments below, organizing them in a sequential order to help their identification in the revised version of the manuscript.

Review report of the paper "Variability and trend analysis of temperature in the upper troposphere and stratosphere region over the tropics (Réunion), by combining balloon-sonde and satellite measurements" by Gregori de Arruda Moreira et al. The paper "Variability and trend analysis of temperature in the upper troposphere and stratosphere region over the tropics (Réunion), by combining balloon-sonde and satellite measurements", authors investigated how temperature and the height of the tropopause are changing over the tropics near Réunion Island using 15-year data from radiosonde, COSMIC-1 and MERRA-2. Using the combined datasets, they assess how well different methods match when estimating the tropopause height to explore a refined long-term trend. Studying tropopause temperature and height variability is very important for understanding the anthropogenic contribution to climate change, which is a highly relevant topic in ANGO. Given the sparse availability of datasets from the Southern Hemisphere, a long-term report on this subject from Réunion is particularly valuable and much needed. However, the referee feels that the paper currently lacks some important details, and the authors should provide further clarification on following concerns before the manuscript can be considered ready for acceptance.

**Major concerns:**

1. Methodology: The methodology section requires more detail and clarification. The authors state that they used radiosonde temperature profiles from 2006–2020 (available weekly) together with COSMIC-1 RO data within a margin of ±2° and ±3° spatial resolution. However, it is not clearly described how simultaneous these datasets were. Although they showed examples of simultaneous measurements in Figure 2, a clear description should be given in the methodology. As in Table 1 indicates a larger number of COSMIC profiles, but it is not specified whether the authors computed weekly averages of these data to match the radiosonde records, or how day-to-day variability was accounted for. Although a long-term analysis, a clear description of data is required for the reliability of the derived tropopause characteristics.

The final dataset was created by combining SHADOZ data (1 profile per week) and COSMIC-1 data (1 profile approximately every 2 days). For days when data from both instruments coincided, priority was given to using SHADOZ data. In order to clarify this point in the main text, this phrase has been added:

Line 310:

*"SHADOZ and COSMIC-1 have different temporal resolutions (1 profile per week and 1 profile every 2 days, respectively), as mentioned in section 4.2. Therefore, the final database was created from the combination of these two datasets, and for days where there is data from both instruments, only the SHADOZ data were considered."*

2. Another concern about the Trend-Run model, the authors used a simplified approach to evaluate trend by only reporting the coefficient of determination ($R^2$). However, $R^2$ alone cannot provide information about the reliability or statistical significance. The authors are encouraged to include uncertainty analyses, for example confidence intervals and performing sensitivity tests to demonstrate which forcings have the most significant influence on the results.

Within Trend-Run, we consider that two successive measurements are not necessarily independent. The degree of dependence between successive measurements is evaluated using the autocorrelation coefficient. This coefficient makes it possible to determine the uncertainty on the estimated trend and on the contribution of each forcing. The trend uncertainty therefore, represents the uncertainty in the trend slope value.

The uncertainties associated with the trend estimates and with the contributions of individual forcings are provided in the article. As an example, uncertainties on the forcings are illustrated by the shaded areas in Fig. 2b, while trend uncertainties are shown by the grey shaded region and the dotted lines in Fig. 10.

3. Lack of proper discussion: Using the combined dataset, the authors identified differences/similarities in estimating tropospheric heights both CPT and LRT and reported some seasonal variations and small but significant long-term changes in tropopause temperature and height, which are important for understanding climate change. However, they did not properly discuss these results in the broader context of atmospheric dynamics, radiative forcing, or anthropogenic climate forcing. The discussion is limited to a brief comparison with Sivakumar et al. (2011) and Bègue et al. (2010) etc. The authors should expand the discussion of the results with additional published works and explain how their results align or diverge from global findings, focusing on the uniqueness or similarities of TTL dynamics in the southern hemisphere. Without a clear discussion and interpretation of the results, the manuscript remains incomplete in its current form.

We thank the reviewer for this comment. In order to improve this paper, the following paragraph has been added:

Line 362:

*"Furthermore, it is important to highlight that the observed trends for both the CPT (cooling) and the LRT (warming) temperatures are directly associated with tropospheric warming, which has been exacerbated by intense accumulation of GHG in the lower troposphere as described by Ladstädter et al., 2023. This phenomenon is a key indicator of climate change and has been observed globally (Ladstädter et al., 2025), reinforcing the intense effect of anthropogenic activities across the planet."*

**Minor concerns:**

1. The introduction could be improved by more clearly highlighting key gaps in studying tropopause height and temperature over the tropics, and particularly importance of study over

Réunion. While this section is otherwise well written and easy to follow, motivation of the study would be stronger if the authors emphasized these gaps more explicitly.

In order to solve this question, the following phrases have been added:

Line 42

*"However, they are endowed with lower temporal resolution. Therefore, considering the limitations and advantages of each methodology, an option to improve the tropopause monitoring is to combine them. Although, firstly, it is necessary to identify their similarities and differences."*

2. Line 100: I feel the sentence 'Therefore, temperature gradients ….. and chemistry.' is irrelevant here.

Done